# Best Policy Tracking in Gradient-based Optimization

**Judith Echevarrieta[1], Etor Arza[2], Aritz Pérez[3], Josu Ceberio[1]**
[1]University of the Basque Country UPV/EHU,
{judith.echevarrieta,josu.ceberio}@ehu.eus
[2]Norwegian University of Science and Technology NTNU, etor.arza@ntnu.no
[3]Basque Center for Applied Mathematics BCAM, aperez@bcamath.org

## Abstract

Policy optimization in reinforcement learning consists of optimizing an agent's decision-making strategy, based on experience gained through interaction with an environment and with the goal of best solving the task determined by the environment. Gradient-based algorithms have proven effective by representing the agent's behaviour with stochastic neural network policies. Multiple reinforcement learning libraries have been created to facilitate problem-solving and the development of new algorithms. In experimental studies, these tools are often treated as black boxes, focusing primarily on the final policy returned by the algorithm rather than on understanding how it was chosen from the entire sequence of visited policies. However, gradient-based algorithms suffer from high variance gradient estimates, leading to significant oscillations in the performance of consecutive visited policies. Under this phenomenon, selecting the best policy from the whole sequence of visited policies becomes a critical issue, as naive choices, such as selecting the last policy, might lead to undesired policies and inefficient learning time investment. This project aims to investigate the relevance of this problem. To that end, we will examine the limitations of existing approaches and will explore if new methods can improve the selection of the best-visited policy.

## 1  Motivation

In Reinforcement Learning (RL), Policy Optimization (PO) consists of finding the optimal decision-making approach to solve a task, based on the experience observed during the interaction [18]. The decision-maker is called the *agent*, and everything outside the agent that makes the interaction possible is called the *environment*. Interaction happens continuously during learning: the agent selects actions and the environment responds to those actions by rewarding them and presenting new states to the agent. This interaction allows the agent to improve its way of acting.

A wide variety of heuristic algorithms have been designed to solve PO problems [2]. They iteratively determine how the agent should update its way of acting by considering the feedback from its interaction with the environment. These algorithms are considered RL algorithms, among which gradient-based PO algorithms, a.k.a. Policy Gradient (PG), stand out [16, 17] for being suitable for learning policies in environments with both discrete and continuous action spaces. PG algorithms define the behaviour of the agent as an explicit *policy*, i.e., a stochastic neural network that maps each state of the environment to a probability distribution over possible actions. The policies allow interaction with the environment for finite periods of time that result in sequences of data formed by states of the environment, actions of the agent and rewards of these actions provided by the environment. Finally, the sequences resulting from the interaction are used to approximate the gradient of the policy performance and apply stochastic gradient descent to update the policy. Agent-environment interaction and policy update are alternated iteratively until the time available for learning is exhausted.

XVI XVI Congreso Español de Metaheurísticas, Algoritmos Evolutivos y Bioinspirados (maeb 2025).

In recent years, many libraries [3, 6, 9, 11, 13, 15] have been developed, aimed at supporting research in RL. They contain implementations of multiple RL environments and algorithms, facilitating experimentation and evaluation of new proposals. Those libraries are used as a baseline to experimentally test the performance of new algorithms [5, 14], as they ease the comparison to existing ones. They are also used to simulate multiple tasks in continuous learning, identifying different tasks with different environments [12]. In these cases, we observe that the RL community uses libraries in the following way: First, an environment and an algorithm are selected from those implemented by the library. Then, a random seed is set, and the algorithm is run in the environment for a certain time, leading to a *learning process*. Finally, the policy that is returned by default is used as the learned policy.

Consequently, the use of RL libraries can be interpreted as a black-box learning process. In fact, experimental works in the literature do not pay much attention to how the returned policy is selected from the sequence of policies generated during learning. However, the implementations of PG algorithms estimate the gradients of policy performance using data sequences gathered during their interaction, which leads to high variance estimates [7, 10]. This may result in unwanted policy updates with significant oscillations in performance. Therefore, the choice of the returned policy can by no means be considered trivial due to the oscillations that appear during the learning process. This project aims to analyze if it is possible to improve the best policy selection.

## 2    Problem Definition and Previous Work

Formally an environment is defined as a Markov Decision Process [18], a tuple $\mathcal{E} = (S, A, p, r)$ where $S$ is the state space, $A$ is the action space, $p \colon S \times A \to \Delta(S)$ is the state transition probability function that defines a probability distribution $p(\cdot|s, a)$ with support in $S$ for a given state-action pair $(s, a) \in S \times A$, and $r \colon S \times A \to \mathbb{R}$ is the reward function. The agent interacts with the environment according to a probability function $\pi \colon S \to \Delta(A)$, called the *policy*, which defines a probability distribution $\pi(\cdot|s)$ with support in $A$ for a given state $s \in S$.

At each step of the interaction, the agent starts from the current state of the environment $s \in S$ and chooses an action $a \sim \pi(\cdot|s)$. Then, the environment rewards the chosen action $r(s, a)$ and updates the state $s' \sim p(\cdot|s, a)$. The agent-environment interaction breaks down intentionally with a stopping criterion (e.g., a number of maximum steps), or naturally upon reaching a special state. These states are *terminal states* $S_T \subset S$, which are reset by a non-terminal state called *initial state* from $S_I \subset S$ before acting, drawn from a probability distribution $p_I \in \Delta(S_I)$. A sequence formed by the state-action pairs $\tau = \big((s_1, a_1), ..., (s_n, a_n)\big)$ generated during a $n$ step interaction of a policy $\pi$ with an environment $\mathcal{E}$ is named *trajectory*. When the first state of a trajectory is initial $s_1 \in S_I$, the last state-action pair leads to a terminal state $s_{n+1} \in S_T$ and the intermediate states are not terminal $s_i \in S \setminus S_I$ for all $i \in \{2, ..., n-1\}$, the trajectory is called an *episode*. Each trajectory is given a reward computed from the rewards of its state-action pairs. The *trayectory reward* is defined as the discounted cumulative reward that, with a slight abuse of notation, we formally denote as

$$r(\tau) = \sum_{i \in \{1, ..., |\tau|\}} \gamma^i r(s_i, a_i) \tag{1}$$

where $\gamma \in [0, 1]$ is the *discount factor* and $|\tau|$ is the length of $\tau$. When $\tau$ is an episode, $r(\tau)$ is called *episodic reward*, $\gamma = 1$ and $|\tau|$ is a priori unknown.

In a PO problem, the objective is to find the policy that maximizes the average episodic reward over randomly initialized episodes [16]. Since policies are stochastic probabilistic models, in a fixed environment starting from the same initial state, the same policy may generate different episodes. Therefore, the performance of a policy $\pi$ in an environment $\mathcal{E}$ is the expected episodic reward

$$f(\pi) = \mathbb{E}_{s_1 \sim p_I} \mathbb{E}_{\tau \sim \pi \times p|s_1} r(\tau) \tag{2}$$

where $\tau \sim \pi \times p|s_1$ denotes an episode with initial state $s_1$, generated with the policy $\pi$ and the transition distribution $p$ of the environment $\mathcal{E}$. Consequently, the PO problem is formulated as an optimization problem

$$\pi^* = \arg\max_{\pi \in \Pi} f(\pi) \tag{3}$$

where $\Pi$ is the space of all possible policies.

The PG algorithms optimize Problem (3) using stochastic gradient descent [7]. The gradient $\nabla f(\pi)$ is approximated from the trajectory rewards of the finite number of trajectories generated during the

previous interaction of $\pi$ with the environment $\mathcal{E}$, and the policy is updated according to this estimate. The agent-environment interaction and policy update are iteratively repeated until the maximum available time $t_{max}$ for learning (e.g., maximum number of iterations) is exhausted. This results in a sequence of policies $(\pi_1, ..., \pi_{t_{max}})$ that completely determines the learning process defined by the environment, the PG algorithm and the random seed.

The drawback of the PG algorithms when solving a PO problem is that gradient estimators may have a high variance [2, 7, 10]. This sometimes causes a *degradation* in learning, i.e., updates to policies with an expected episodic reward significantly lower than that of previous policies. Under the degradation phenomenon, an inappropriate selection of the returned policy from the sequence of policies generated during learning may have undesirable consequences. We have reviewed some state-of-the-art RL libraries [11, 13, 15], and we observe that the implementations on RL libraries consider three simple criteria for the selection of the output policy, each one with its pros and cons:

**Last visited policy**. Libraries such as Stable Baselines3 [15] by default return the last policy of the sequence of policies. Despite the simplicity of this criterion, its effectiveness is highly dependent on the level of degradation as shown in Figure 1. This selection criterion can lead to potentially undesirable policies under critical or catastrophic degradations, as well as to an inefficient time investment. For instance, in the last two graphs of Figure 1 the last policy has a very similar performance to the first one, which has been chosen randomly.

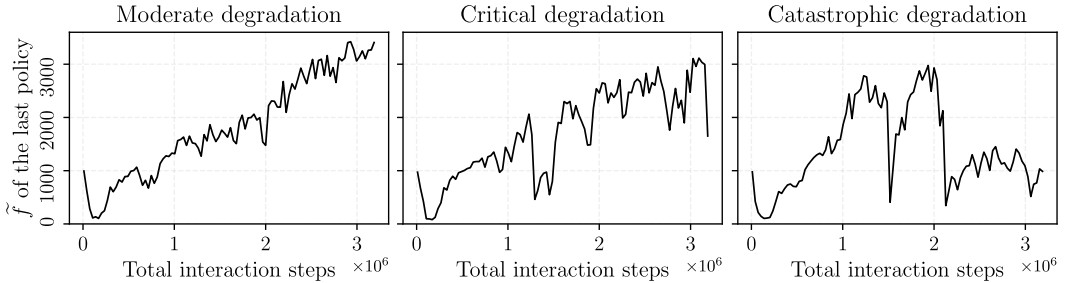

Figure 1: Learning curves with different levels of degradation for Ant environment and PPO algorithms implemented in Stable Baselines3 [15], and three seeds. The curves show the evolution of $\widetilde{f}(\pi_t)$ for $t \in \{1, ..., t_{max}\}$, where $\widetilde{f}(\pi_t)$ is the average episodic reward over 100 randomly initialized episodes generated sequentially with $\pi_t$, $t_{max} = 3.2 \cdot 10^6$ steps and $n = 2048$ steps per interaction.

**Best policy using training data**. The best policy of the sequence of policies is returned, according to an estimate of the expected episodic reward using the trajectories stored during learning interactions. For example, Sample Factory [13] and RL Games [11] evaluate the updated policies with a periodic frequency, considering the average episodic reward over a finite number of previous episodes completed during the interaction. Finally, the selected policy is the one among the evaluated ones with the highest value of that metric. The more training episodes that are used, the more data will have been generated by previous policies, and therefore different from the current policy. This results in a trade-off between the number of episodes considered for the estimation and the level of representation of these data concerning the policy to be evaluated, as shown in the left graph of Figure 2.

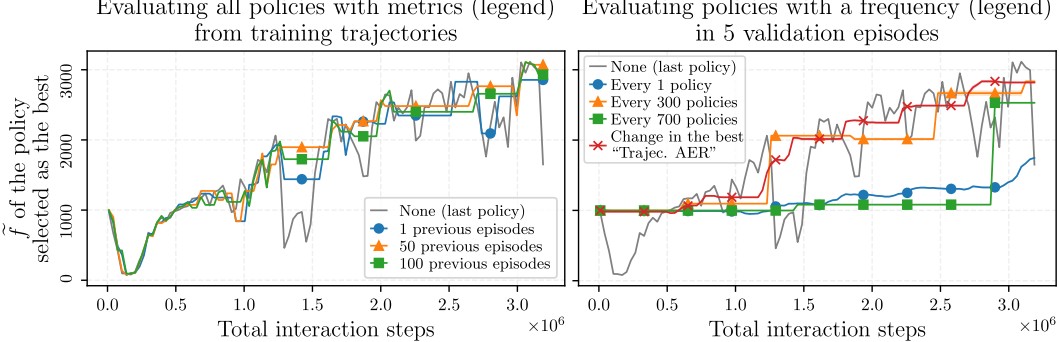

Figure 2: Learning curves obtained after selecting the best policy with the techniques considered in the state-of-the-art (with —, ●, ▲ and ■) or a suggested improved technique (with ×).

**Best policy using validation data**. The best policy of the sequence is returned, in terms of an estimation from additional validation episodes generated in an independent interaction. For instance, Stable Baselines3 allows evaluating periodically the performance of the policies in a finite number of additional episodes to those of the training trajectories, and finally returns the policy with the highest average episodic reward. This technique considers only data generated by the policies themselves for evaluation, but by considering an additional interaction, it decreases the remaining time for learning. The higher the frequency of evaluation, the less the extra overhead, but the more likely we are to overlook a good policy. Therefore, a trade-off between validation frequency (time spent on validation) and the probability of detecting the best policy appears, as shown in the right graph of Figure 2.

## 3 Hypothesis

Considering the simplicity, efficiency and effectiveness of the existing techniques, the starting hypothesis in this work is: *the development of new methods based on the combination of the good characteristics of the existing ones could improve the tracking of the best policy among the sequence of updated policies*. If correct, this would improve the solution for the *Best Policy Tracking* problem

$$\pi_t^* = \arg\max_{\pi \in (\pi_1, \ldots, \pi_t)} f(\pi), \quad \text{for all } t \leq t_{max}. \tag{4}$$

which is the anytime version of Problem (3). A preliminary experiment to motivate our hypothesis is illustrated by the red curve with cross markers of the right graph in Figure 2. It is the learning curve obtained after defining the validation frequency from changes detected in the average episodic reward (AER) of the training trajectories generated by each policy. Therefore, although validation takes additional time, performing it only when a change is detected in the best-updated policy seems to potentially ensure more effective tracking of the best policy.

## 4 Objectives and Methodology

To validate our hypothesis, we will focus on answering the following question: how relevant is the selection of the best policy among the sequence generated during RL process beyond existing research?. Aligned to that, we will address two objectives described below:

**Objective 1.** We will start analysing the existing techniques by answering: can the validation frequency and the number of validation episodes be improved?. For this purpose, we will first study the similarity between policies and their performances, since the frequency may be set on the assumption that it is not necessary to evaluate all policies because nearby policies will have similar performances. Second, we will compare how costly it is to validate a policy in a sample of episodes versus the time consumed in the learning interaction. With parallel execution, it may be possible to evaluate a sufficiently large number of episodes to distinguish correctly between policies while consuming only a small percentage of the interaction time. In this case, setting a fixed number of validation episodes would be unnecessary.

**Objective 2.** We try to improve the techniques proposed in the state-of-the-art by answering: can we combine the training and validation estimates of the expected episodic reward to obtain a better trade-off between validation cost and policy tracking efficiency?. First, we will propose a new method that can combine the good characteristics of the existing ones. Then, we will test the new proposal in different environments and algorithms, and compare it with the existing methods. This comparative experiment will allow us to assess the difficulty of designing a technique that can work well across the many different possibilities for formulating and solving a PO problem. If the number of RL environments and algorithms with which a single method can appropriately track the best policy is irrelevant, the development of a new method would be senseless.

The described methodology will allow us to conclude the veracity of our hypothesis, thus if it is indeed possible to develop new methods that better track the best policy on multiple PO problems. If correct, the proposed method itself, which affirms the hypothesis, will serve not only to improve existing methods, but will be more generalizable and favour early stopping in learning tasks that are no longer interesting to invest more time in. This may have interesting applications in multi-task learning or continuous learning [1, 12], or even in single-task learning to distinguish between less and more promising runs differentiated by the random seed [4, 8].

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
