# OpenReview forum: "Best Policy Tracking in Gradient-based Optimization"
_MAEB/2025/Projects_Track — MAEB 2025 Proyectos_

### Official Review · Reviewer_nVc7 · 2025-03-18
**Proyecto interesante y bien descrito**

**Rating:** 5
**Confidence:** 3

**Review:**

El proyecto se centra en el contexto de agentes y la obtención de políticas de decisión para resolver problemas específicos. Las posibles políticas se obtienen mediante algoritmos de aprendizaje por refuerzo.
La pregunta clave que plantean es ¿qué relevancia tiene la selección de la mejor política entre la secuencia generada durante el proceso de aprendizaje?
El proyecto presentado define bien los objetivos y muestra algunos resultados preliminares.
Sería interesante explorar posibles conexiones con el campo de optimización robusta, donde se busca que la solución obtenida se comporte "bien" en escenarios similares (pero no exactamente iguales) que el utilizado en la fase de optimización.
También se podrían explorar otras definiciones para "best policy" que incluyeran (en el momento de seleccionarla) otros aspectos como la complejidad computacional, "explicabilidad", etc.

---

### Official Review · Reviewer_ecaS · 2025-03-18
**Presentación de un proyecto interesante**

**Rating:** 5
**Confidence:** 4

**Review:**

Este trabajo presenta lo que previsiblemente es un proyecto de investigación enfocado al estudio del Policy Optimization problem. Este problema se centra en la selección de la mejor política para un agente que se desenvuelve en un entorno.

El trabajo presenta de una manera muy clara el problema que se pretende abordar, se proporciona una descripción matemática de dicho problema y se describen los principales enfoques en los que está trabajando la comunidad científica. Seguidamente, se plantea la hipótesis principal del trabajo: la elaboración de nuevos métodos de selección de políticas basados en las mejores características de los métodos utilizados por la comunidad científica. Finalmente, se proporcionan los objetivos del proyecto.

Como digo, el trabajo parece bastante interesante y prometedor, y por ello, no veo razones para rechazar este trabajo. Estaré pendiente a los avances en este área.

---

### Official Review · Reviewer_rZFN · 2025-03-19
**This project proposal addresses policy optimization in reinforcement learning, examining existing limitations and exploring new methods,.**

**Rating:** 4
**Confidence:** 4

**Review:**

In this project proposal, the authors address the policy optimization problem in reinforcement learning. To this end, they examine the limitations of existing approaches and explore new methods to tackle the problem.

The proposal is reasonably interesting, well-written, and easy to follow, making it suitable for further development. Additionally, the motivation for the project is supported by a preliminary experiment.

The main flaw of the proposal is the lack of clarity regarding its relation to heuristics/metaheuristics. In other words, it is unclear how the techniques, either proposed in the state of the art or in this project, are relevant to the audience of a heuristics/metaheuristics conference. This should be explained and clarified.

Additionally, there are a few minor aspects related to the organization of the proposal that should be taken into consideration:

- The title of Section 2 is misleading. You named it "Hypothesis," but it contains the definition of the problem, the main techniques, and only a few lines at the very end stating the hypothesis. Please consider rewording and splitting the section into "Problem Definition," "Previous Methods," and "Hypothesis." In fact, the hypothesis could be stated at the end of the introduction.
- In Section 3, objectives and methodology should be separated.

---

### Decision · Program_Chairs · 2025-03-19

Accept